# Design and Fabrication of Embedded Microchannel Cooling Solutions for High-Power-Density Semiconductor Devices

**DOI:** 10.3390/mi16080908

**Published:** 2025-08-04

**Authors:** Yu Fu, Guangbao Shan, Xiaofei Zhang, Lizheng Zhao, Yintang Yang

**Affiliations:** 1School of Microelectronics, Xidian University, Xi’an 710126, China; cyberfu@126.com (Y.F.); ytyang@xidian.edu.cn (Y.Y.); 2China Aerospace Institute of Standardization, Beijing 100071, China; 3National Innovation Institute of Defense Technology, Academy of Military Sciences, Beijing 100071, China; zxf1990fei@163.com; 4The 12th Research Institute of China Electronics Technology Group Corporation, Beijing 100071, China; larazlz@163.com

**Keywords:** semiconductor devices, thermal management, microchannel, micropillar array, near-junction

## Abstract

The rapid development of high-power-density semiconductor devices has rendered conventional thermal management techniques inadequate for handling their extreme heat fluxes. This manuscript presents and implements an embedded microchannel cooling solution for such devices. By directly integrating micropillar arrays within the near-junction region of the substrate, efficient forced convection and flow boiling mechanisms are achieved. Finite element analysis was first employed to conduct thermo–fluid–structure simulations of micropillar arrays with different geometries. Subsequently, based on our simulation results, a complete multilayer microstructure fabrication process was developed and integrated, including critical steps such as deep reactive ion etching (DRIE), surface hydrophilic/hydrophobic functionalization, and gold–stannum (Au-Sn) eutectic bonding. Finally, an experimental test platform was established to systematically evaluate the thermal performance of the fabricated devices under heat fluxes of up to 1200 W/cm^2^. Our experimental results demonstrate that this solution effectively maintains the device operating temperature at 46.7 °C, achieving a mere 27.9 K temperature rise and exhibiting exceptional thermal management capabilities. This manuscript provides a feasible, efficient technical pathway for addressing extreme heat dissipation challenges in next-generation electronic devices, while offering notable references in structural design, micro/nanofabrication, and experimental validation for related fields.

## 1. Introduction

With the rapid advancement of semiconductor technology, electronic devices are evolving toward higher power densities, increased integration levels, and smaller dimensions [1]. From central processing units (CPUs) and graphics processing units (GPUs) in high-performance computing (HPC) to widely adopted semiconductor power devices in 5G communications and electric vehicles, heat generation per unit area has surged dramatically. Heat flux densities now commonly exceed hundreds of W/cm^2^ [2,3]. This trend has created critical “thermal bottleneck” issues: excessively high junction temperatures not only severely constrain device performance and operational speed but also accelerate material degradation, significantly compromise long-term reliability, and may even trigger catastrophic thermal failures [4,5]. Consequently, developing efficient and compact thermal management technologies has become one of the critical challenges for advancing modern electronics.

Conventional thermal management approaches, such as forced air cooling and standard macro-scale liquid cooling, struggle to meet the cooling demands of contemporary high-heat-flux devices due to limitations in heat transfer coefficients and dissipation capacities [6]. To address this challenge, significant attention has been directed toward microchannel cooling technology since its initial proposal by Tuckerman and Pease in the 1980s [7]. This approach creates microscale channels directly within or in the near-junction region of device substrates, employing forced convection of coolants (e.g., deionized water, refrigerants, or liquid metals) for heat removal. Leveraging exceptionally high surface area to volume ratios, microchannels achieve substantially greater heat transfer efficiency than conventional methods and are recognized as one of the most promising solutions for high-heat-flux thermal management [8,9].

Over the past few decades, microchannel cooling technology has been extensively investigated by researchers. Regarding structural design, studies have progressed from initial straight channels to various complex topologies, including spiral, serpentine, fractal networks, and manifold configurations. These developments aim to enhance fluid distribution uniformity, reduce pressure drops, and improve overall thermal performance [10,11]. In terms of heat transfer mechanisms, single-phase liquid cooling has been widely examined for its operational stability and reliability, while two-phase flow boiling cooling demonstrates significant potential for high-heat-flux scenarios by utilizing phase-change latent heat to achieve superior heat transfer coefficients—though challenges like flow instabilities persist [12]. Concurrently, substantial research has been conducted on microchannel fabrication techniques [13] and practical implementation strategies [14]. Despite considerable progress, significant challenges remain in seamlessly integrating efficient microchannel cooling solutions with actual high-power-density semiconductor devices and achieving thermal management under extreme heat fluxes exceeding 1000 W/cm^2^. Notably, most current research focuses on optimizing individual aspects, lacking a comprehensive and systematic approach that spans structural design, numerical simulation, advanced manufacturing, and experimental validation. Additionally, effectively regulating flow boiling behavior at the microscale to maximize heat transfer benefits while ensuring system stability represents a critical unresolved scientific challenge.

Therefore, an embedded microchannel cooling solution tailored for high-power-density devices is proposed in this manuscript. Initially, a heat dissipation structure based on a micropillar array was designed and optimized through finite element simulation, with the performance of micropillars of various geometries being compared. Subsequently, a comprehensive multilayer microstructure fabrication process was developed, incorporating key techniques such as deep reactive ion etching (DRIE) on silicon substrates, surface hydrophilic/hydrophobic modification, and gold-silicon eutectic bonding. Finally, an experimental test platform was established to comprehensively assess the thermal performance of the fabricated device under an elevated heat flux of up to 1200 W/cm^2^, where the maximum device temperature during operation reached only 46.7 °C. This work not only demonstrates an advanced cooling technology capable of effectively meeting extreme thermal management requirements but also provides viable technical pathways for thermal design and integration in future high-performance electronic devices.

## 2. Related Work

Microchannel cooling technology is a heat dissipation method that involves designing microscale channels within substrates to remove heat via fluid flow [15]. Its advantage lies in the high surface-area-to-volume ratio, significantly enhancing convective heat transfer efficiency, making it particularly suitable for high-power-density devices. The dimensions of microchannels are typically in the order of micrometers, and decreasing channel dimensions generally increases the convection heat transfer coefficient and cooling efficiency. Though microchannel cooling faces challenges such as ensuring fluid flow uniformity and mitigating channel clogging, these issues can be resolved through optimized design and material selection.

Regarding different microchannel structures and topologies, Lanying Zhang and Yangfei Zhang [16,17] employed the finite volume method to investigate cooling microchannels integrated within Low-Temperature Co-fired Ceramic (LTCC) substrates and their heat transfer characteristics. This study examined straight, I-shaped, and spiral microchannels, demonstrating spiral channels’ superior cooling capability with maximum temperatures reduced to 74.84 °C. Alena Pietrikova and Tomas Girasek et al. [18] researched four distinct microchannel configurations for chip packaging, utilizing simulation software to analyze thermal resistance, flow paths, coolant distribution, and thermal profiles in LTCC substrates with embedded microchannels. The findings reveal significant impacts of channel geometry, dimensions, and positioning on thermal resistance, coolant pressure, and cooling chip efficiency. The study further examined volume flow effects on substrate thermal resistance and thermal cross-talk between chips. Hejie Yu and Min Miao et al. [19] implemented finite element modeling for double-layer spiral and serpentine microchannel topologies, concluding that cooling capacity primarily depends on temperature rise in the fluid-outlet substrate layer. Kevin P. Drummond and Justin A. Weibel et al. [20] studied samples featuring different channel lengths (750 and 1500 μm) and depths (125, 250 and 1000 μm), with constant 60 μm channel and fin widths. Liquid coolant (HFE-7100) was delivered at consistent flow rates with inlet velocities around 1.05 m/s. Visualization of two-phase flow was achieved using high-speed cameras through glass sidewalls, while infrared cameras measured temperature distributions along channel walls.

For microchannel fabrication methods, Tong et al. [21] bonded 0.7-mm-diameter copper microtubes to ceramic substrate backplanes using SAC305 solder, followed by encapsulation with high-power light-emitting diode modules. This approach reduced chip surface temperature with increasing coolant flow rates, though further velocity increases yielded diminishing cooling improvements. He et al. [22] implemented manifold microchannels and junction cooling techniques in AlN ceramic substrates, forming a three-dimensionally heterogeneously integrated chip. Experimental results demonstrated exceptional thermal dissipation of 700 W/cm^2^ at standard operating temperatures. Furthermore, sensitivity analysis and multi-objective optimization through the NSGA-II algorithm optimized manifold microchannels within the chip, reducing total thermal resistance by 13.6% and maximum pressure drop by 68.5%. These studies form a theoretical framework for high-power integrated chips, advancing the development of high-performance integrated circuits. Bohan Tian and Deming Yang et al. [23] proposed a fabrication method for alumina heat spreaders embedded with oscillating heat pipes, effectively preventing crack formation while achieving sealed interconnected microchannels within the substrate. Mikulics et al. [24] demonstrated a breakthrough in thermal management for AlGaN/GaN heterostructures by utilizing metallic silver substrates. Through high-resolution Raman thermography, their study revealed a 60% reduction in channel temperature (at 7 W/mm power dissipation) compared to conventional sapphire substrates. This enhanced heat dissipation was attributed to silver’s superior thermal conductivity, which decreases by only ~5% in the operational temperature range (300–700 K), contrasting sharply with the ~30% drop observed in SiC. Recent advancements in interface-engineering strategies, including surface roughening, functional interlayers, and diffusion-barrier coatings—have demonstrated significant potential in reducing interfacial electrical resistance and enhancing mechanical bond strength for thermoelectric devices. As highlighted in the roadmap by Wu and Liu [25], such approaches are critical for mitigating elemental interdiffusion and chemical degradation at electrode interfaces. Notably, their work underscores the efficacy of phase-diagram-driven design in achieving ultra-stable contact resistivity (R_c_ < 1 mΩ cm^2^) under prolonged thermal stress.

Furthermore, liquid materials for microchannels have been extensively investigated. Nian Liu and Min Miao [26] employed low-melting-point metal alloys as the coolant medium while incorporating through-holes in the microchannel region to enhance thermal dissipation. A three-dimensional numerical simulation was performed to comparatively analyze heat transfer performance between the metal alloy and deionized water under identical dynamic conditions and varying heat flux densities. The results indicate that deionized water exhibits inadequate cooling when heat flux reaches 5 W/cm^2^. At a flow rate of 70 mL/min using the low-melting-point alloy, the substrate’s peak temperature was maintained at 354 K under 10 W/cm^2^ heat flux. With further heat flux escalation to 30 W/cm^2^, combining the alloy flow at 70 mL/min with through-hole implementation reduced maximum temperatures below 357 K. Rui Zhang and Marc Hodes et al. [27] tested microchannel heat sinks using single-phase water and liquid metal under high heat flux conditions. Flow rates and thermal resistance data were recorded under distinct parametric conditions, reproducing thermal resistance values reported by Tuckerman and Pease while achieving a higher heat flux of 835 W/cm^2^ under identical parameters. Thermal stress reliability remains a key research focus. Wei He et al. [28] established a finite element model for substrates with embedded microchannels to analyze thermal stress–strain distributions. Single-factor analysis on straight-channel configurations revealed that maximum substrate stress decreases with increasing channel cross-sectional dimensions but varies with inter-channel spacing. Additionally, a finite element model for Ball Grid Array (BGA) solder joints on microchannel ceramic substrates was developed, coupled with fluid–structure interaction simulations [29]. Three microchannel topologies (straight, square-loop, and serpentine) and their geometric parameters were evaluated for their impact on solder joint stress. The results showed that the microchannel spacing has the greatest influence on the stress of BGA solder joints, followed by the length and width of the microchannel cross-section.

This study designs micropillar microchannel structures and liquid distribution networks through finite element simulations, developing and implementing multilayer microfluidic fabrication processes. Liquid forced-convection cooling directly applied to electronic devices is complemented by controlled flow boiling operations, maintaining optimal heat transfer efficiency within the nucleate boiling regime, ultimately achieving a cooling capacity of 1200 W/cm^2^.

## 3. Design, Simulation, and Manufacturing

### 3.1. Numerical Modeling and Design

Based on finite element theory, embedded cooling microchannel structures within high-power chip substrates were designed using COMSOL Multiphysics (version 6.1). Heat transfer characteristics from the junction region to the substrate were specifically investigated, focusing on embedded channel structures with high contact surface area and low flow resistance. Particular emphasis was placed on studying the microfluidic delivery volume under high-heat-flux dissipation requirements while considering the thermophysical properties of cooling fluids.

Structures for micropillar cooling channels and liquid distribution networks were separately designed and simulated. During the forced convective two-phase cooling process in micropillar configurations, the structure of micropillars significantly influences boiling behavior and heat exchange performance. Therefore, four distinct micropillar geometries were designed for this study, as shown in Figure 1. All structures shared identical characteristic dimensions: micropillar edge length (or diameter) *D* = 60 μm, and inter-pillar pitch *P* = 100 μm.

As shown in Figure 2 and Figure 3, these are schematic diagrams of the bonding profile of the micropillar structure device and the top view of the device structure. During the test, it is necessary to observe and record the two-phase states within the device and characterize the thermal exchange characteristics of the device. Two methods, silicon–glass anode bonding and silicon–silicon bonding, were adopted.

The microchannel liquid separation structure was fabricated from single-crystal silicon wafers using MEMS processing techniques, with overall dimensions standardized at 6.0 mm × 7.0 mm to maintain precise dimensional compatibility with the co-packaged GaN High Electron Mobility Transistor (HEMT) semiconductor die. This configuration enables direct hydraulic integration through which liquid coolant is routed into parallel embedded microchannels and subsequently collected following heat exchange processes. To ensure uniform coolant distribution across the microfluidic network, the inlet and outlet manifolds were designed with square cross-sections (500 × 500 μm), precisely patterned through photolithographic processes. Two distinct flow path topologies were implemented: a baseline single-inlet/single-outlet configuration and an enhanced single-inlet/dual-outlet arrangement with bifurcated flow paths, as shown in Figure 4.

### 3.2. Fabrication Process

The multilayer microchannel structure requires separate process development for its three components: embedded microchannels (primary cooling section), coolant inlet/outlet ports, and an intermediate liquid distribution layer adapted to chip cooling requirements.

#### 3.2.1. Development of Microchannel Cooling Structures

The microchannel fabrication process comprised thermal oxidation, photolithography, RIE etching, and deep silicon etching. Specific procedures are detailed in Table 1, with preliminary etch results shown in the micrograph of Figure 5.

#### 3.2.2. Development of the Coolant Distribution Structure

The primary layer constitutes the coolant distribution structure, schematically illustrated in Figure 6a,b, fabricated using two photomasks. This structure incorporates four inlets, with etched channels on Surface A dividing the incoming coolant into four streams. The secondary layer functions as the coolant reflow structure, channeling the coolant toward two outlets. Its schematic and fabrication process are depicted in Figure 6c,d, also requiring two photomasks.

#### 3.2.3. Development of the Surface Structure Modulation Process

For flow boiling heat dissipation in microchannels, rapid boiling initiation and efficient bubble transport should be achieved. The microchannel walls must be treated to form a combination of smooth hydrophilic and hydrophobic regions. Hydrophobic zones facilitate rapid boiling, while hydrophilic regions enable rapid heat transfer and bubble removal so that the flow path remains unobstructed by generated bubbles. A microscopic composite thin film deposition and etching process is employed to fabricate surface structures with varied roughness levels, achieving targeted control of microchannel walls. Figure 7 demonstrates different contact angles corresponding to three surface roughness levels.

#### 3.2.4. Sealing and Bonding

Conventional structural joining techniques, including epoxy adhesion, thermal compression, welding, or thermal grease bonding, typically introduce substantial thermal interface layers between heat sources and cooling layers. These layers obstruct rapid vertical heat transfer and diffusion. Thus, low-thermal-resistance bonding technology is developed to address multilayer stacking issues in thermal management.

Given the current design, the fluid velocity in the cooling layer sufficiently covers the entire microfluidic chamber. Considering the liquid inlet/outlet diameter of Φ2 mm, the maximum flow rate does not exceed 7 mL/s. This flow requirement can be met with standard miniature liquid-cooling pumps. At this flow rate, the fluid pressure within the cooling layer remains below 500 kPa—significantly less than the Au-Sn bonding strength (50 MPa). Therefore, the Au-Sn bonding solution fully satisfies sealing requirements.

### 3.3. Experimental Setup and Testing Procedure

The experimental system incorporates a deionized water tank, a liquid pump, microscopic imaging equipment, K-type thermocouples with digital displays, a power supply for heating elements, and connecting tubing. A photograph of the experimental test platform is shown in Figure 8.

Data collected include liquid temperatures at the cooler inlet/outlet, heating element temperature, input power to heating elements, and visual observation of two-phase heat dissipation. Two GaN PA chip samples with MEMS microchannel coolers were tested as follows: (a) The pump was activated to gradually increase the flow rate to 100 mL/min until the fluidic system stabilized; (b) power was applied to GaN PA Chip 1 at 0.5 W, with infrared thermography identifying the maximum temperature as Measurement Point 1. After power-off, this process was repeated for Chip 2 to establish Measurement Point 2; (c) with GaN PA Chip 1 powered to target settings, voltage (U) and current (I) values were recorded after thermal stabilization; (d) temperature T_1_ was measured at Point 1 using infrared thermography; and (e) after power shutdown, steps (c–d) were repeated for Chip 2 to record T_2_. The relevant information of the proposed GaN PA chip can be found in the team’s published study [30].

## 4. Results and Discussion

### 4.1. Analysis of Simulation Results

After completing the structural design, the forced convection cooling performance of the micropillar array was simulated for single-phase operation using COMSOL Multiphysics (version 6.1). Figure 9 shows the model of the square micropillar heat sink, with pillar dimensions of 60 μm × 60 μm × 250 μm (width × length × height), inter-pillar pitch *P* = 100 μm, and a heated area of 0.5 cm × 0.5 cm.

Monocrystalline silicon was defined as the solid material, with water as the coolant.

The multiphysics simulation configuration established the following boundary conditions: ambient temperature T_0_ = 20 °C; the coolant inlet volumetric flow rate is 120 mL/min; the outlet pressure is configured as 122 kPa with backflow suppression enabled to ensure unidirectional flow; and heated surface power density q = 500 W/cm.

This setup simulated subcooled convective heat transfer under pressurized conditions. The outlet pressure exceeded atmospheric pressure to prevent cavitation, while the flow rate ensured turbulent flow for enhanced heat exchange efficiency. The elevated power density emulated extreme thermal loads typical of high-power electronic devices.

The temperature distribution from the simulation is presented in Figure 10. The maximum temperature of 97.6 °C occurred at the downstream region of the heated surface. The average outlet fluid temperature reached 62.347 °C, while the overall average temperature was 41.019 °C.

The actual heat transfer coefficient h of the microstructure was calculated using Equation (1):(1)$$h=\frac{P_{\mathrm{in}\;}\times A_{\mathrm{heat}\_\mathrm{area}}}{\left(T_{\mathrm{avg}\_\mathrm{chip}}-T_{\mathrm{avg}\_\mathrm{water}}\right)\times A_{\mathrm{heat}\_\mathrm{exchange}\_\mathrm{area}}}$$
where *P*_in_ is the input thermal power, *A*_heat_area_ is the heat source area, *T*_avg_chip_ and *T*_avg_water_ are average temperatures of the solid domain and liquid domain, respectively; finally, *A*_heat_exchange_area_ represents the heat exchange area. The simulated results yielded a heat transfer coefficient of 1240 W/(m^2^·K).

### 4.2. Experimental Performance Evaluation

Following the experimental platform and procedures described in Section 3.3, two GaN power amplifier chips from China Electronics Technology Group Corporation (Beijing, China) of comparable dimensions were selected as heat sources. As shown in Figure 11, each chip measured 1.044 mm × 0.779 mm, with a gate region dimension of 0.744 mm × 0.204 mm.

DC power was supplied separately to each chip. Voltage was adjusted until the chips reached thermal equilibrium at an operating power of 1.823 W, corresponding to a heat flux density of 1200 W/cm^2^. The temperature distribution was recorded using an infrared thermal imager. The maximum temperatures for Chip 1 and Chip 2 were measured as 46.7 °C and 40.6 °C, respectively (Table 2, Figure 12 and Figure 13).

Our experimental results demonstrate that the junction-adjacent cooling technology effectively dissipated extreme heat fluxes. By directly etching microchannels onto the backside of high-power devices, the thermal path between heat sources and coolant was shortened, thereby reducing thermal resistance and enabling efficient heat transfer. At 1200 W/cm^2^ heat flux, the maximum device temperature was maintained at 46.7 °C using this microchannel structure, confirming its superior cooling capability.

To further evaluate the advancement of this work, the traditional methods and implementation methods of microchannels with different design structures, such as manifold and porous interconnected, and manufacturing materials such as copper and AlGaN were selected for comparison. The results in Table 3 show that the present square micropillar design with gold-tin bonding demonstrated 1200 W/cm^2^ heat dissipation at only 27.9 K temperature rise (18.7 °C baseline), exhibiting significant superiority.

## 5. Conclusions

An embedded microchannel cooling solution for high-power-density semiconductor devices was successfully designed, fabricated, and validated. Through systematic integration of finite element simulation, advanced micro-nanofabrication, and comprehensive experimental characterization, the solution demonstrated significant potential for extreme thermal management challenges. The proposed micropillar-array embedded cooler effectively maintained the maximum GaN device temperature at 46.7 °C under 1200 W/cm^2^ heat flux, corresponding to a 27.9 K temperature rise. This performance surpasses multiple state-of-the-art microchannel solutions referenced in the literature, proving the efficacy of junction-adjacent direct cooling strategies. Future work will focus on the following: investigating two-phase flow boiling mechanisms in microstructures for enhanced heat transfer coefficients through surface engineering and channel optimization and conducting long-term reliability studies to establish foundations for commercial deployment.

## Figures and Tables

**Figure 1 micromachines-16-00908-f001:**
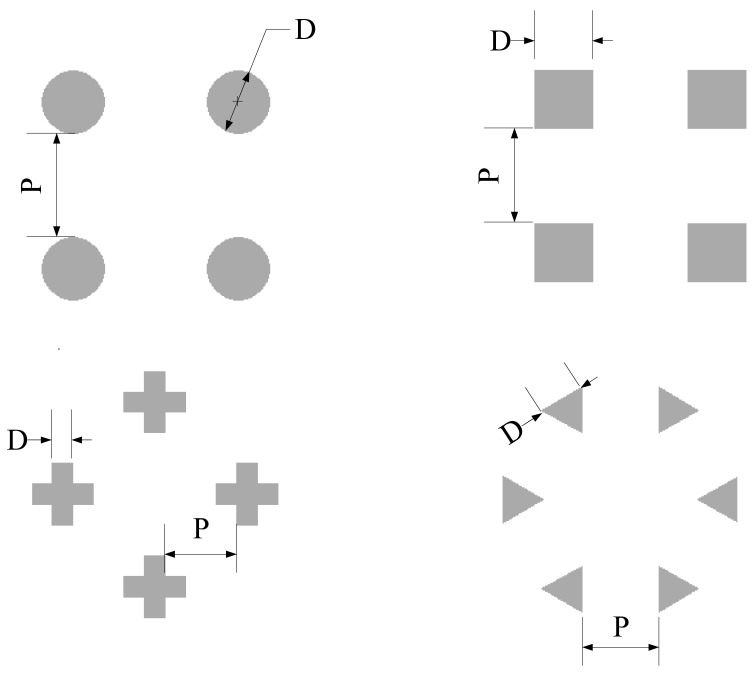
Four different micropillar structures: circle, square, cross, and triangle.

**Figure 2 micromachines-16-00908-f002:**
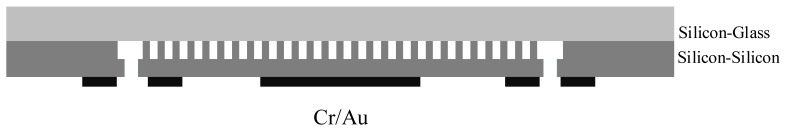
Schematic diagram of the bonding profile of the micropillar structure.

**Figure 3 micromachines-16-00908-f003:**
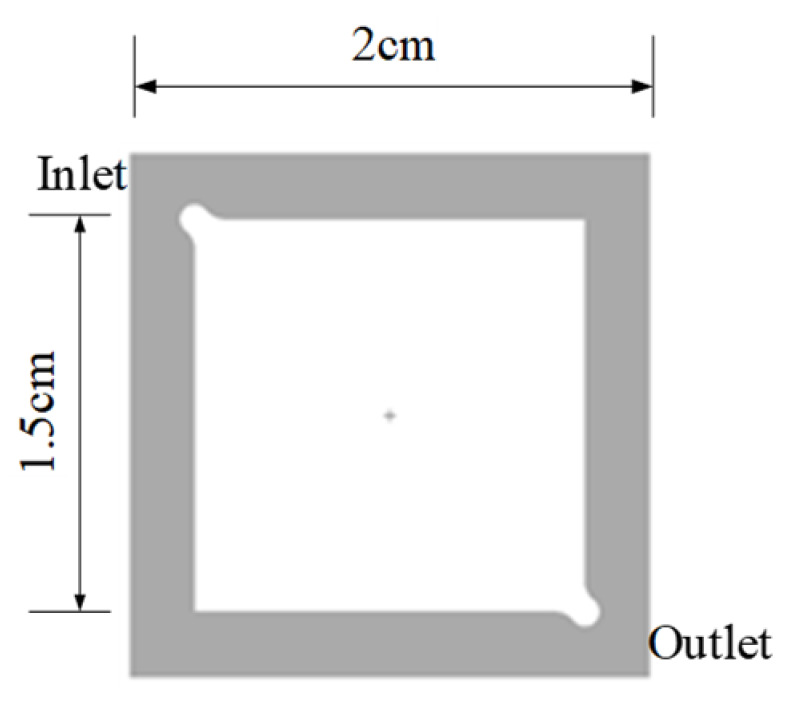
Top view and dimensions of the structure.

**Figure 4 micromachines-16-00908-f004:**
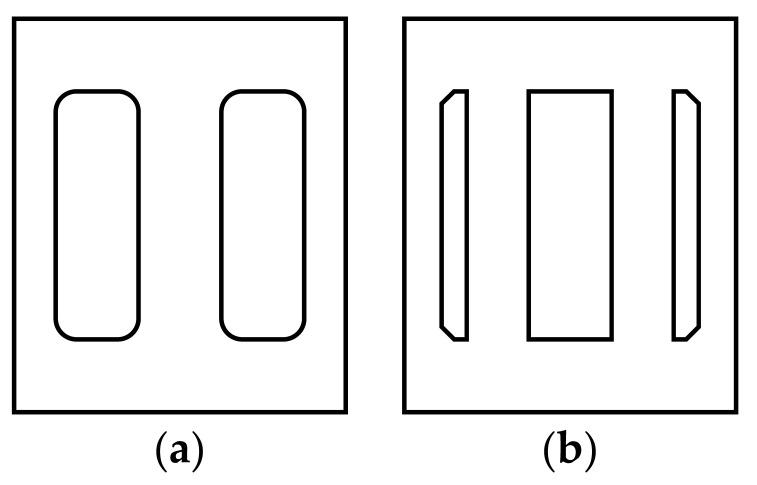
Two kinds of liquid separation structures: (**a**) single-in single-out structure and (**b**) single-in double-out structure.

**Figure 5 micromachines-16-00908-f005:**
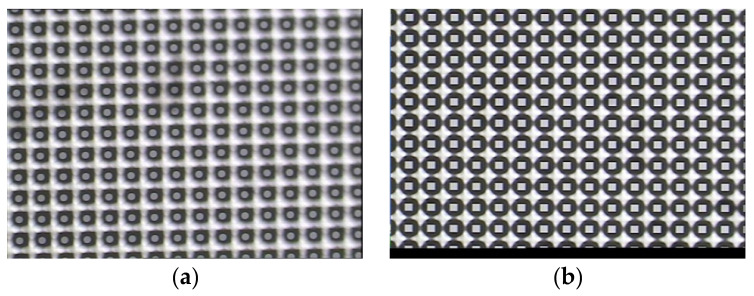
Micrographs of etch results: (**a**) circle, (**b**) square, (**c**) cross, and (**d**) triangle.

**Figure 6 micromachines-16-00908-f006:**
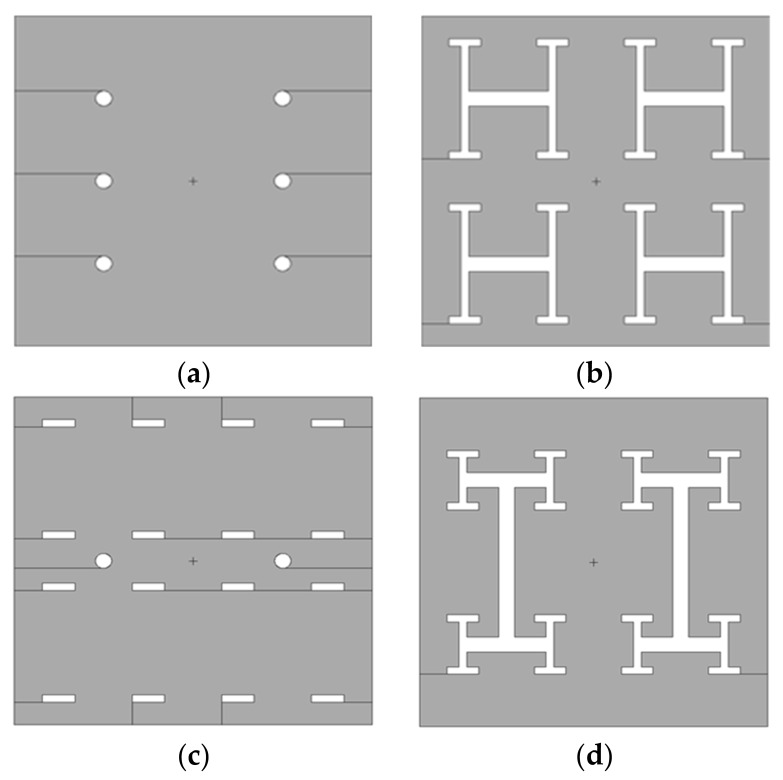
Fabrication schematics of coolant distribution and reflow structures: (**a**,**b**) represent the deep silicon etching masks for Surfaces A and B of the coolant distribution structure; (**c**,**d**) show the deep silicon etching masks for Surfaces A and B of the coolant reflow structure.

**Figure 7 micromachines-16-00908-f007:**
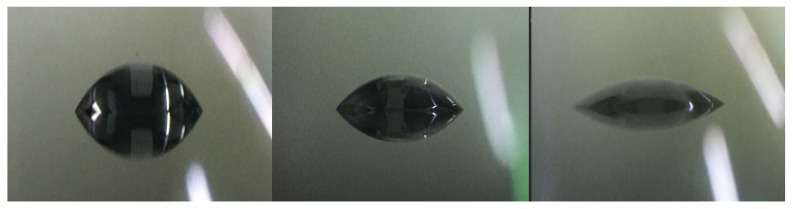
Three types of contact angle effects.

**Figure 8 micromachines-16-00908-f008:**
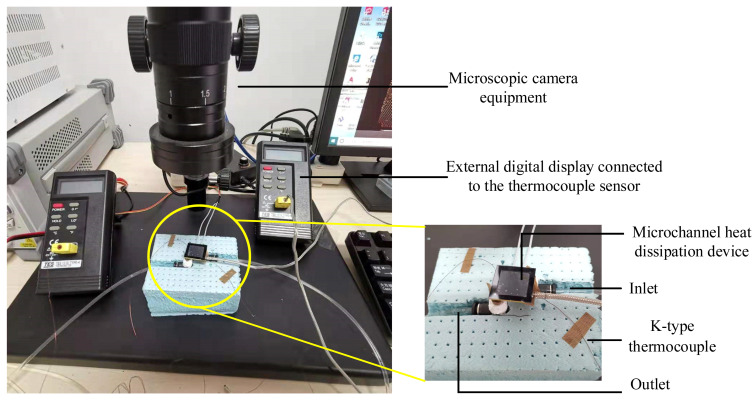
Experimental test platform.

**Figure 9 micromachines-16-00908-f009:**
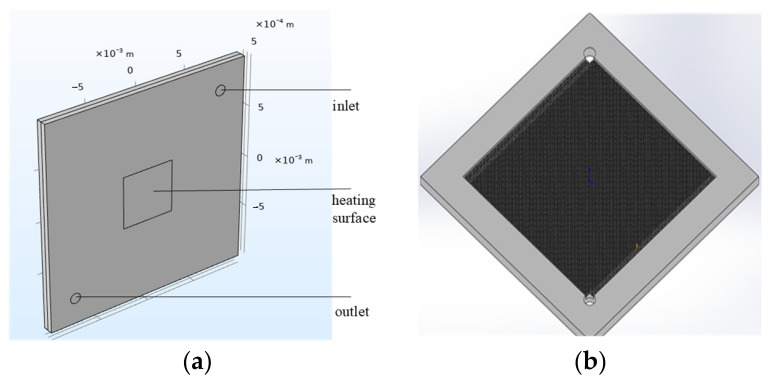
Square micropillar cooler model: (**a**) overall structural schematic and (**b**) internal configuration of cooler.

**Figure 10 micromachines-16-00908-f010:**
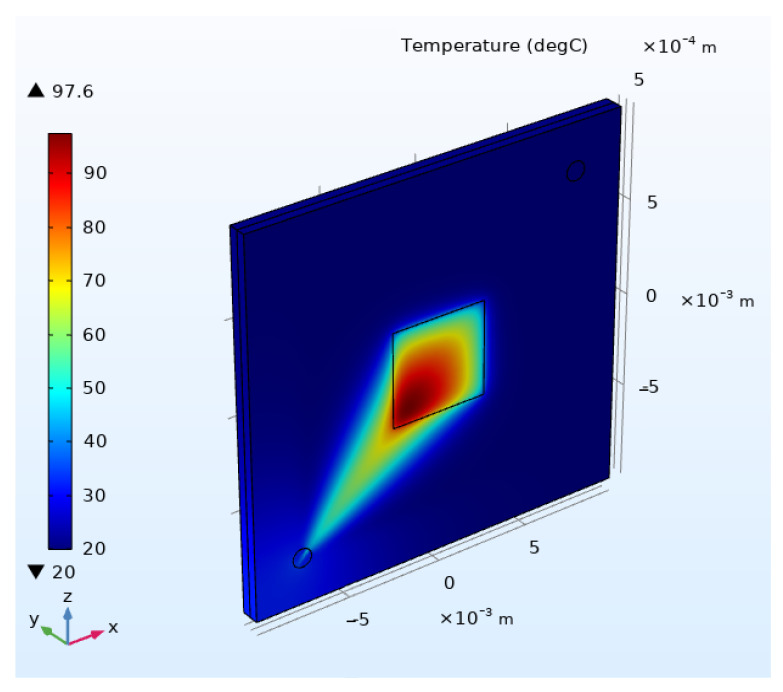
Simulated temperature field distribution.

**Figure 11 micromachines-16-00908-f011:**
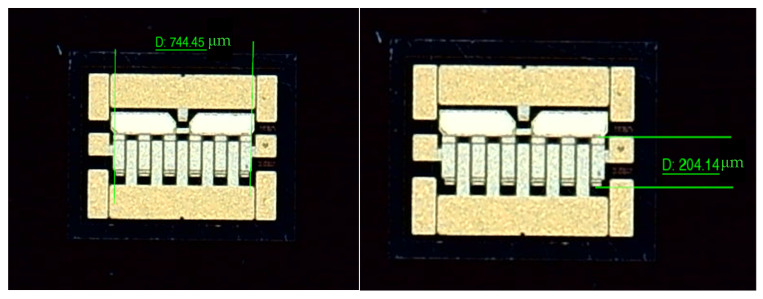
Dimensions of the GaN PA chip.

**Figure 12 micromachines-16-00908-f012:**
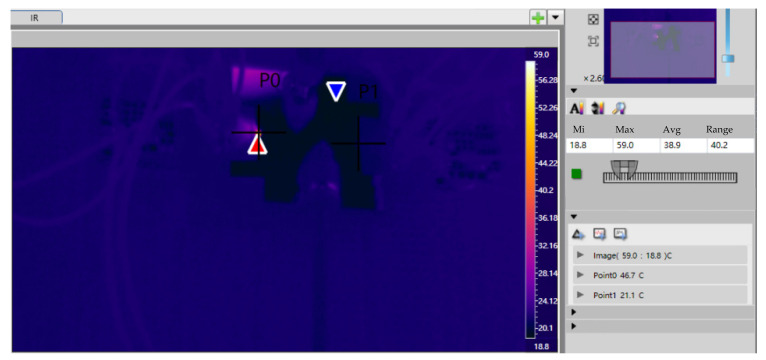
Thermal test record of Chip 1 for the microchannel sample. The positions of the red arrows and the blue arrows are the high-temperature and low-temperature zones respectively.

**Figure 13 micromachines-16-00908-f013:**
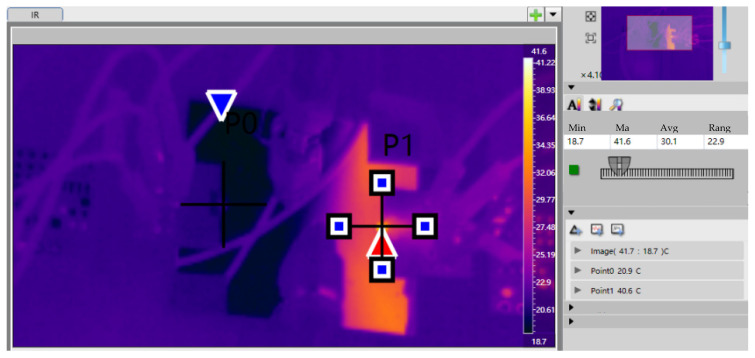
Thermal test record of Chip 2 for the microchannel sample. The positions of the red arrows and the blue arrows are the high-temperature and low-temperature zones respectively.

**Table 1 micromachines-16-00908-t001:** Fabrication process steps and parameters.

Step	Process	Parameters
1	Substrate preparation	4-inch *P* <100>, 1–10 Ω∙cm, 525 µm thickness, double-side polished
2	Thermal oxidation	100 ± 10 nm (hard mask)
3	Backside photolithography	Inlet/outlet patterning
4	RIE oxide etching	Over-etching
5	Deep silicon etching	260 ± 10 µm
6	Frontside photolithography	Micropillar patterning
7	RIE oxide etching	Over-etching
8	Deep silicon etching	260 ± 10 µm
9	Bonding	Channel sealing

This table records the key steps and parameters of the manufacturing process.

**Table 2 micromachines-16-00908-t002:** Thermal performance test records of microchannel samples.

Sample	Power (W)	Current (A)	Voltage (V)	Temperature (°C)
Chip 1	1.823	0.606	3.009	46.7
Chip 2	1.823	0.554	3.303	40.6

The room temperature is 18.7 °C.

**Table 3 micromachines-16-00908-t003:** Comparison of heat dissipation efficiency with similar works.

Works	Dissipation Method	Structure	Material	Heat Flux (W/cm^2^)	Temperature Rise (K)
[31]	Heat sink	SiP	PCM	20 W/-	77
[32]	Forced air cooled	Fin	PCM	20 W/-	31.9
[22]	Microchannel	Manifold	Copper	700	-
[33]	Microchannel	Porous interconnected	Copper	200~500	16.7
[34]	Microchannel	Porous-fin	Porous copper	100	-
[35]	Microchannel	Parallel and manifold	AlGaN and GaN	1000 (single-phase water-cooling)	60
[36]	Microchannel	Nanoporous membrane	SOI and silicon	(665 ± 74)	(28.5 ± 1.8)
Our work	Microchannel	Square microcolumn	Silicon	1200	27.9

This table shows the comparison between the microchannel structure implemented in this study and traditional heat dissipation methods, as well as other similar works. PCM is a kind of material: phase change material.

## Data Availability

The original contributions presented in the study are included in the article, further inquiries can be directed to the corresponding author.

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
