# Peer review of "Design and Fabrication of Embedded Microchannel Cooling Solutions for High-Power-Density Semiconductor Devices"

_micromachines, 2025, doi:10.3390/mi16080908_

Round 1

Reviewer 1 Report

Comments and Suggestions for Authors

In the manuscript entitled: “Design and Fabrication of Embedded Microchannel Cooling Solutions for High-Power Density Semiconductor Devices” the authors report on device improvements by a new technological approach resulting in a significant temperature decrease in the presented high-power semiconductor devices.

The presented results confirm that the use of the embedded microchannel cooling solution—by directly integrating micropillar arrays within the near-junction region of the substrate—results in effective heat dissipation. This novel-demonstrated solution maintained the maximum GaN device work temperature at 46.7°C under 1200 W/cm² heat flux, corresponding to a 27.9 K temperature rise.

This presented device fabrication procedure represents a significant step forward in device development, especially for high-power applications.

This is an excellent systematic study, which contains new and important information for a broad “high-power devices” and especially HEMT community and material science readership. The results presented in this work really impressed me, and I strongly recommend this work for publication in Micromachines after a minor revision.

I have only several questions, remarks, and/or suggestions listed in the following:

Regarding the presented/used “GaN PA chip samples with MEMS microchannel coolers” in this study, could you please implement information about the heterostructure—both channel and barrier material? If you cannot present these results in this present manuscript, please implement them in your future/further publications. Thanks in advance.

Furthermore, for the sake of comparison, it would be very appreciated by the “high-power device community” if you could compare device characteristics for both “conventional” and “improved” (with microchannel cooling solution) devices. Explicitly, it would be great if you could present comparatively at least static transfer and transconductance characteristics for the presented HEMT. Again, if you cannot present these results in this present manuscript, please implement them in your future/further publications. Again, thanks in advance!

Please implement into the manuscript information about the threshold voltage for HEMT devices presented in this study.

Next, did you perform any long-term reliability measurements on your presented “improved” devices for, let’s say, 10 or 100 hours of operation?

For the sake of comparison, the authors should report in the introduction section additional related work focused on the use of “metallic” substrates, explicitly direct epitaxial growth, e.g., “GaN” or III-nitrides for high-power device applications, for the sake of improved heat management. Although you performed broad research on different solutions related to improving thermal heating management, the following work should be cited in any case.

[1] Mikulics, M.; Kordoš, P.; Fox, A.; Kočan, M.; Lüth, H.; Sofer, Z.; Hardtdegen, H. Efficient heat dissipation in AlGaN/GaN heterostructure grown on silver substrate. Appl. Mater. Today 2017, 7, 134–137.

Author Response

Comments 1: Regarding the presented/used “GaN PA chip samples with MEMS microchannel coolers” in this study, could you please implement information about the heterostructure—both channel and barrier material? If you cannot present these results in this present manuscript, please implement them in your future/further publications.

Response 1: Thank you for pointing this out. We agree with this comment. Details regarding the GaN PA chip were previously published in another study by our team (reference 30 on page 14). To avoid redundancy, we have intentionally omitted these technical specifics in the current manuscript.

Comments 2: Furthermore, for the sake of comparison, it would be very appreciated by the “high-power device community” if you could compare device characteristics for both “conventional” and “improved” (with microchannel cooling solution) devices. Explicitly, it would be great if you could present comparatively at least static transfer and transconductance characteristics for the presented HEMT. Again, if you cannot present these results in this present manuscript, please implement them in your future/further publications.

Response 2: Thank you for pointing this out. We agree with this comment. We have now incorporated traditional air-cooling thermal management data into Table 3 for comparative analysis. This extended version is presented on page 11.

Comments 3: Please implement into the manuscript information about the threshold voltage for HEMT devices presented in this study.

Response 3: Thank you for pointing this out. We agree with this comment. Details regarding the GaN PA chip were previously published in another study by our team (reference 30 on page 14). To avoid redundancy, we have intentionally omitted these technical specifics in the current manuscript.

Comments 4: Next, did you perform any long-term reliability measurements on your presented “improved” devices for, let’s say, 10 or 100 hours of operation?

Response 4: Thank you for this valuable observation. Long-term reliability testing has not yet been conducted. The reviewer’s suggestion is highly appreciated and will be prioritized in future studies.

Comments 5: For the sake of comparison, the authors should report in the introduction section additional related work focused on the use of “metallic” substrates, explicitly direct epitaxial growth, e.g., “GaN” or III-nitrides for high-power device applications, for the sake of improved heat management. Although you performed broad research on different solutions related to improving thermal heating management, the following work should be cited in any case.

Response 5: Agree. Expanded discussions on prior work—particularly regarding "metal" substrates—have been incorporated into the Introduction (page 3, line 134).

Reviewer 2 Report

Comments and Suggestions for Authors

This manuscript makes a significant contribution by integrating finite-element-driven design of micropillar geometries, a robust multilayer MEMS fabrication process (DRIE, wettability patterning, Au–Sn bonding), and comprehensive two-phase boiling experiments at heat fluxes up to 1200 W/cm². The work demonstrates a remarkable 27.9 K temperature rise and outperforms existing microchannel solutions, offering a clear, scalable pathway for next-generation electronic cooling.I recommend acceptance pending minor revisions to address the above comments.

  1. the manuscript thoughtfully introduces four distinct micropillar geometries and emphasizes controlled flow boiling, the current presentation focuses exclusively on square pillars and evaluates performance almost entirely through temperature metrics. To reinforce the rationale for selecting the square design, it would be beneficial to supplement the study with comparative simulations of thermal resistance, heat transfer coefficients, and pressure drop for the circular, cross-shaped, and triangular pillars, alongside a quantitative examination of bubble nucleation, growth, and departure in each configuration. Likewise, enriching the boiling-mechanism discussion by including high-speed imaging or cross-sectional micrographs that capture bubble dynamics, reporting critical heat flux and two-phase heat transfer coefficients under comparable conditions, and addressing potential flow instabilities—such as vapor backflow or vapor trapping—and their effects on stability and performance would significantly strengthen the paper’s mechanistic insights and overall impact.
  2. While the manuscript demonstrates effective multilayer integration via Au–Sn eutectic bonding, it does not quantify the interfacial thermal resistance at the bond interface or consider the electrical contact resistance that may evolve during thermal cycling. Mechanical reliability issues arising from coefficient‐of‐thermal‐expansion mismatch at the micropillar/substrate and solder interfaces—such as delamination or void formation under repeated heat loads—are also unaddressed. Moreover, the Introduction and Discussion lack a survey of advanced interface‐engineering strategies (e.g., surface roughening, functional interlayers, diffusion‐barrier coatings) shown to reduce interfacial resistance and enhance bond strength. Incorporating a brief review of these methods—such as (J. Materiomics 10 (3), 748–750, 2024)—would enrich the discussion of device reliability and performance.
  3. Additionally, please ensure that figures and tables are cited in strict numerical order, with each caption clearly defining axes, units, and measurement directions, and that in-text reference labels (e.g., “[22]”, “[28]”) precisely match the bibliography. Standardize terminology by using “Au–Sn eutectic bonding” throughout, correct grammatical slips (for example, replace “are schematic diagram” with “is a schematic diagram”), and adopt consistent unit symbols (“µm” rather than “μm”, “W/cm²” rather than “W/cm2”). Finally, add units to all table column headers (e.g., Heat flux [W/cm²]; Temperature rise [K]) and include a brief explanatory note under each table summarizing its key parameters for improved reader clarity.

Author Response

Comments 1: the manuscript thoughtfully introduces four distinct micropillar geometries and emphasizes controlled flow boiling, the current presentation focuses exclusively on square pillars and evaluates performance almost entirely through temperature metrics. To reinforce the rationale for selecting the square design, it would be beneficial to supplement the study with comparative simulations of thermal resistance, heat transfer coefficients, and pressure drop for the circular, cross-shaped, and triangular pillars, alongside a quantitative examination of bubble nucleation, growth, and departure in each configuration. Likewise, enriching the boiling-mechanism discussion by including high-speed imaging or cross-sectional micrographs that capture bubble dynamics, reporting critical heat flux and two-phase heat transfer coefficients under comparable conditions, and addressing potential flow instabilities—such as vapor backflow or vapor trapping—and their effects on stability and performance would significantly strengthen the paper’s mechanistic insights and overall impact.

Response 1: Thank you for pointing this out. Micro-pillar geometries fall beyond this manuscript’s scope, as parametric simulations of thermal resistance, heat transfer coefficients, and pressure drop present significant time constraints. Quantitative analysis of bubble nucleation, growth, and departure similarly requires dedicated investigation. Current equipment limitations preclude high-speed flow visualization within the revision period. However, the reviewer’s suggestions are highly valuable and will inform future dedicated studies.

Comments 2: While the manuscript demonstrates effective multilayer integration via Au–Sn eutectic bonding, it does not quantify the interfacial thermal resistance at the bond interface or consider the electrical contact resistance that may evolve during thermal cycling. Mechanical reliability issues arising from coefficient‐of‐thermal‐expansion mismatch at the micropillar/substrate and solder interfaces—such as delamination or void formation under repeated heat loads—are also unaddressed. Moreover, the Introduction and Discussion lack a survey of advanced interface‐engineering strategies (e.g., surface roughening, functional interlayers, diffusion‐barrier coatings) shown to reduce interfacial resistance and enhance bond strength. Incorporating a brief review of these methods—such as (J. Materiomics 10 (3), 748–750, 2024)—would enrich the discussion of device reliability and performance.

Response 2: Thank you for pointing this out. We agree with this comment. The effective interfacial thermal resistance and reliability of Au–Sn eutectic bonding layers have been extensively documented in our prior publication (reference 30 on page 14), and are thus omitted here to avoid redundancy. Meanwhile, advanced microchannel fabrication strategies have been expanded in the Introduction (page 3, line 141).

Comments 3: Additionally, please ensure that figures and tables are cited in strict numerical order, with each caption clearly defining axes, units, and measurement directions, and that in-text reference labels (e.g., “[22]”, “[28]”) precisely match the bibliography. Standardize terminology by using “Au–Sn eutectic bonding” throughout, correct grammatical slips (for example, replace “are schematic diagram” with “is a schematic diagram”), and adopt consistent unit symbols (“µm” rather than “μm”, “W/cm²” rather than “W/cm2”). Finally, add units to all table column headers (e.g., Heat flux [W/cm²]; Temperature rise [K]) and include a brief explanatory note under each table summarizing its key parameters for improved reader clarity.

Thank you for pointing this out. We agree with this comment. We have thoroughly examined the citation sequence and correspondence of all charts and references, further checked the standardized terms and unified the expressions throughout the text, corrected grammatical errors as much as possible, and finally added units in all table and column headings and brief annotations. For details, please refer to Tables 1 to 3.

Reviewer 3 Report

Comments and Suggestions for Authors

please consider the following:

- some acronyms seem not to be explained at their first use, such as LTCC, BGA or HEMT. Although some of them are well known for scientists working in the field, but   providing the full versions would greatly improve the paper accessibility,

- the numerical software used for FEM simulations should be mentioned already in Section 2.1. Is it the Comsol  Multiphysics mentioned in Section 4.1?

-  Figure 1 should be placed after it is referred in the text,

- Table 1 and Figure 5 should be swapped,

- some figures, such as Figure 6 or 9b, lack dimensions,

- it would be nice to have a 3D view of the whole system combining the structures from Figures 3-7,

- it makes no sense to provide the value of the heat transfer coefficient in line 300 with such a high accuracy, 1240 would be fine,

- Table 3 has no/wrong caption and some words, such as square or Porous start with capital letters,

- the last column in Table 3 could be replaced with the value of total thermal resistance from junction to ambient, it would be better for solution comparison purposes since it is more power independent.

Author Response

Comments 1: some acronyms seem not to be explained at their first use, such as LTCC, BGA or HEMT. Although some of them are well known for scientists working in the field, but   providing the full versions would greatly improve the paper accessibility,

Response 1: Thank you for pointing this out. We agree with this comment.  We have added explanations at the positions where the relevant terms first appear. For details, please refer to line 102, 168, 184 on page 3-4.

Comments 2: the numerical software used for FEM simulations should be mentioned already in Section 2.1. Is it the Comsol Multiphysics mentioned in Section 4.1?

Response 2: Thank you for pointing this out. We agree with this comment.  We have added the specific names of FEM simulations in Section 2.1. See line 184 on page 4.

Comments 3:  Figure 1 should be placed after it is referred in the text,

Response 3:hank you for pointing this out. We agree with this comment. We have modified the position of Figure 1 in the manuscript. For details, see line 198 on page 5.

Comments 4: Table 1 and Figure 5 should be swapped,

Response 4: Thank you for pointing this out. We agree with this comment.  We have swapped the positions of Table 1 and Figure 5. For details, please refer to page 6 of the manuscript.

Comments 5: some figures, such as Figure 6 or 9b, lack dimensions,

Response 5: Thank you for pointing this out. Figure 6 is only a schematic diagram. Its dimensional information is subject to the actual object. The dimensional information in Figure 9b is consistent with that in Figure 9a.

Comments 6: it would be nice to have a 3D view of the whole system combining the structures from Figures 3-7,

Response 6 Thank you for pointing this out. 3D view of the whole system is shown as Figure 9 of the manuscript.

Comments 7: it makes no sense to provide the value of the heat transfer coefficient in line 300 with such a high accuracy, 1240 would be fine,

Response 7: Thank you for pointing this out. We agree with this comment. We have corrected the value of the heat transfer coefficient. For details, please refer to line 315 on page 10 of the manuscript.

Comments 8: Table 3 has no/wrong caption and some words, such as square or Porous start with capital letters,

Response 8: Thank you for pointing this out. We agree with this comment. We have corrected the incorrect parts in the table. For details, please refer to Table3 on page 11 of the manuscript.

Comments 9: the last column in Table 3 could be replaced with the value of total thermal resistance from junction to ambient, it would be better for solution comparison purposes since it is more power independent.

Response 9: Thank you for pointing this out. According to the reviewers' comments, the relevant information of total thermal resistance has been presented in another article published by the team. For details, please refer to literature 30. Since the relevant content has been published, it is not convenient to display it again in the manuscript.